# Assessing the Psychometric Properties of the Illness Management and Recovery Scale: A Systematic Review Using the Consensus-Based Standards for the Selection of Health Measurement Instruments (COSMIN)

**DOI:** 10.3390/bs14040340

**Published:** 2024-04-18

**Authors:** Nuria Martín-Ordiales, María Dolores Hidalgo, María Pilar Martín-Chaparro, Júlia Ballester-Plané, Maite Barrios

**Affiliations:** 1Department of Psychiatry and Social Psychology, University of Murcia, 30100 Murcia, Spain; nuria.martin@um.es (N.M.-O.); mpmartin@um.es (M.P.M.-C.); 2Department Basic Psychology and Methodology, University of Murcia, 30100 Murcia, Spain; mdhidalg@um.es (M.D.H.); jballesterp897@uao.es (J.B.-P.); 3Departament of Psicology, University Abat Oliba CEU, CEU Universities, 08022 Barcelona, Spain; 4Institute of Neurociencies, University of Barcelona, 08035 Barcelona, Spain; 5Institute of Research Sant Joan de Déu, 08950 Barcelona, Spain; 6Department of Social Psychology and Quantitative Psychology, University of Barcelona, 08035 Barcelona, Spain

**Keywords:** recovery, illness management, IMR-S, systematic review, COSMIN

## Abstract

The Illness Management and Recovery Scale (IMR-S) is based on the IMR program, developed to assess the recovery process for people with severe mental disorders by considering the perceptions of clients and clinicians involved in it. The aim of this study was to analyze the psychometric properties of the IMR-S so as to determine the reliability and suitability of its scores for evaluating recovery. Two coders searched five databases for studies, published between January 2004 and May 2023, that describe the psychometric assessment of the IMR-S. Studies were assessed using the COnsensus-based Standards for the selection of health Measurement INstruments (COSMIN) Risk of Bias checklist. Finally, 46 papers were included. Methodological quality was very good for most of the studies that provide information on internal validity, and limited for those that report on responsiveness. Measurement properties were positive for convergent validity and measurement error. The quality of evidence was high for structural validity studies. Although this study only includes research published in English and may have overlooked certain psychometric properties evaluated in studies published in other languages, our findings suggest that the IMR-S is a valid and reliable instrument, demonstrating its potential to offer guidance for clinical practice.

## 1. Background

In recent decades there has been a shift in intervention strategies with people with severe mental disorder, and the importance of incorporating the perspective of the affected person is now recognized [1]. A key milestone in this process was the emergence in the 1980s of the recovery model as a new paradigm in mental health care [2]. According to Roosenschoon et al. (2021) [3], recovery can be understood and analyzed from three perspectives: clinical, focused on symptom reduction; objective, defined on the basis of social functionality and vocation; and subjective, which encompasses concepts such as connectedness, hope, identity, meaning in life, and empowerment. Developing and implementing interventions based on this model can complement traditional treatments and skill-training programs [4].

The Illness Management and Recovery (IMR) program [5], based on five self-management strategies, was developed as a new model of intervention. It consists of five modules: psychoeducation, improvement in the use of medication, relapse prevention, social skills, and coping strategies [6]. The IMR program aims to foster cooperation between clinicians and affected individuals [7,8]. 

To assess the scope of the program, the Illness Management and Recovery Scale (IMR-S) was developed in two versions, one for clinicians, the other for clients. Furthermore, to ensure a comprehensive evaluation of the intervention’s various aspects, items were generated by both IMR clients and clinicians [5]. The goal was to provide insight into how the recovery process is perceived by both [9]. These scales have proven useful for assessing recovery following a variety of therapeutic interventions, evaluating three key components: coping with the disease, knowledge and personal goals, and effective use of medication [10]. These scales are based on the stress vulnerability model of severe mental illness, suggesting that its effects can be mitigated through improved coping skills, social support, and involvement in meaningful activities. 

The IMR-S appears as a strong scale because it assesses domains related to personal recovery, does not exceed 50 items, considers users’ perspectives, presents quantitative data and is scientifically tested [11]. Moreover, some psychometric studies [12] provide information about reliability, validity and responsiveness. Compared with other recovery scales, such as the Recovery Assessment Scale (RAS) [13] or Mental Health Recovery Measure (MHRM) [14], the IMR-S appears to have exhibited the highest level of service user participation throughout its development, providing added value to the scale by enhancing its understandability and applicability [12].

While the IMR-S has been widely applied in the study of recovery in mental health [6,15,16,17,18,19,20,21], the quality of these studies has not, to the best of our knowledge, been examined through a systematic review using the COnsensus-based Standards for the selection of health Measurement INstruments (COSMIN) Risk of Bias checklist [22]. 

COSMIN is a ten-step methodology designed to guide researchers in conducting systematic reviews of patient-reported outcome measures (PROM). The first part guides researchers though performing the literature search (steps 1 to 4). The second part concerns the evaluation of the measurement properties (steps 5 to 7), where the guide provides information on how to evaluate methodological quality and measurement properties using quality criteria, and how to summarize the evidence and grade the quality of evidence. It consists of 10 boxes, containing standards for assessing the methodological quality of studies, including content validity, structural validity, internal consistency, cross-cultural validity, reliability, measurement error, criterion validity, hypothesis testing, and responsiveness. The third part focuses on evaluating the interpretability and feasibility of the PROMs, and formulating and reporting the systematic review (steps 8 to 10). This methodology has established specific and comprehensive guidelines for evaluating content validity; updated criteria for good measurement properties, a risk of bias checklist, an approach for grading the quality of evidence, and synthesizing the overall rating; and formulated recommendation standards for selection of patient-reported outcome measures (PROMs) [22,23,24]. For these reasons, to create a thorough quality assessment of IMR-S, we carried out a systematic review using the COSMIN approach.

The principal aim of this systematic review was to assess the psychometric properties of the IMR-S as reported in studies using either version of the scale (clinician or client), and based on the data extracted to determine the reliability and suitability of IMR-S scores for evaluating recovery. A secondary objective was to analyze the methodological quality of the studies selected for this review.

## 2. Methods

Studies describing the measurement properties of the IMR-S were examined using the latest version of COSMIN, the COSMIN Risk of Bias checklist [22], and the 2020 guidelines for Preferred Reporting Items for Systematic reviews and Meta-Analyses (PRISMA) [25]. The protocol of the current review was registered with the International Prospective Register of Systematic Reviews (PROSPERO 2022 CRD42022354851) in August 2022.

### 2.1. Search Strategy

Five databases (Web of Science, Scopus, PubMed, CINAHL, PsycINFO) were systematically searched for journal articles published from January 2004 through to May 2023. The search terms used were “Illness Management and Recovery” AND “scale”. The search strategy and the general characteristics of the selected studies are included as a Appendix A.

### 2.2. Inclusion and Exclusion Criteria

The inclusion criteria were empirical studies that (1) applied the IMR-S (either the client, clinician, or both versions), (2) were written in English, and (3) reported data on at least one psychometric property as defined in the COSMIN Risk of Bias checklist. Therefore, theoretical papers, systematic reviews, letters, abstracts, protocols, single case studies, and editorials were excluded. Studies that only used a limited number of IMR items, not corresponding to a specific subscale or factor, were also excluded.

### 2.3. Screening

The identified studies were examined by two independent coders. One of the coders was a clinical psychologist with more than five years of clinical experience, while the other was a psychologist with over five years of experience in psychological research. Both coders independently assessed the titles and abstracts of the selected studies. Any discrepancies that arose during the process were discussed through consensus meetings. In the event that uncertainty or disagreement persisted, additional team members were consulted to reach a common decision. Full-text review of the initial sample led to additional articles being excluded. Once again, any differences of opinion were resolved through discussion. The use of two independent coders throughout the study screening process could have helped reduce the number of relevant unidentified studies. However, no computerized information extraction program was used in the process. 

### 2.4. Study Selection and Data Extraction

Eligible studies were examined by the two coders, who extracted pertinent information regarding their general characteristics, as well as indications from the COSMIN Risk of Bias checklist. The general characteristics of studies that were recorded were title, authors, publication year, journal, type of study, sample characteristics, version of the scale applied, number of items, and psychometric data. The COSMIN checklist was used to evaluate the methodological quality and to obtain an overall rating of the scale in terms of its measurement properties.

### 2.5. Evaluation of Selected Studies

The selected studies were analyzed according to the three-phase approach outlined in the COSMIN guidelines [22,24]: (a)Methodological Quality Assessment: As a first step, the methodological quality of the included studies was evaluated using the COSMIN Risk of Bias checklist, which comprises 10 domains describing a specific psychometric property (see Table 1). We systematically assessed each study against the defined criteria outlined in each domain (i.e., box) and assigned a rating of very good, adequate, doubtful, or inadequate methodological quality in each case, based on the degree of rigor in a study’s design and how it was conducted and reported. This phase allowed us to analyze each study individually, understanding the strengths and limitations of the selected studies.(b)Measurement Property Assessment: As a second step, we assessed the measurement properties using predefined quality criteria in alignment with COSMIN guidelines. Each psychometric property was rated as sufficient (+), insufficient (−), or indeterminate (?), based on the applied quality criteria. This phase allowed us to assess the quality of each psychometric property of the IMR-S analyzed.(c)Evidence Synthesis and Quality Grading: As a final step, we summarized the evidence, considering both the methodological quality and measurement property assessments, using the Grading of Recommendations Assessment, Development and Evaluation (GRADE) approach [24] to evaluate the quality of evidence, considering factors such as risk of bias, inconsistency, indirectness, imprecision, and publication bias. The resulting quality level of the evidence in each domain was categorized as high, moderate, low, or very low, indicating the confidence in the results and the trustworthiness of the scale. This phase allowed us to obtain a comprehensive assessment of the evidence studied.

## 3. Results

### 3.1. Study Characteristics

The search process returned a total of 720 reports after removing duplicates. Upon screening titles and abstracts, 567 articles appeared potentially relevant and were retrieved for full-text review. Of these, 510 articles were excluded for various reasons: 39 addressed recovery in other areas of health and did not investigate the targeted outcomes, 84 were other types of study (e.g., protocol, essay), 3 did not apply the entire scale, 39 were reviews, 269 applied other scales, and 3 applied other IMR scales (i.e., Illness Management and Recovery Fidelity scale [15]). Consequently, 46 pertinent studies conducted from 2004 to 2023 were selected for the final analysis. Figure 1 presents a detailed flowchart that outlines the process of article inclusion.

The studies reviewed reported data from both clients and clinicians. The mean number of clients surveyed was 378.2, ranging from 8 [26] to 10,659 users [27]. The mean age of clients was 44.3 years, ranging from 18 to 68.8 years. A total of 17,377 people responded to the scale, of which 7350 were women (42.4%). The studies had been conducted in various countries: USA (*n* = 22), Israel (*n* = 5), Netherlands (*n* = 5), Sweden (*n* = 3), Denmark (*n* = 2), Turkey (*n* = 2), Norway (*n* = 1), Singapore (*n* = 1), England (*n* = 1), Canada (*n* = 1), Kenya (*n* = 1), and Japan (*n* = 1). The IMR-S was applied in different languages, although predominantly in English (n = 30 studies); other languages were Hebrew (**n** = 4), Dutch (*n* = 4), Swedish (*n* = 2), Turkish (*n* = 2), Norwegian (*n* = 1), Arabic (*n* = 1), Danish (*n* = 1), and Swahili and Kamba (*n* = 1). Over time, the number of publications on the IMR-S has shown an upward trend (with the exception of the years affected by COVID-19), and 73.9% of the studies have been published in the last 10 years.

Data showed that 50% of the studies applied only the client version of the IMR-S, 43.3% applied both versions, and 6.5% applied just the clinician version. A significant majority (93.2%) of the studies applied the full scale (15 items). Only 17.4% of reports were psychometric studies (n = 8), while the remaining 82.6% (n = 38) were intervention, case–control or exploratory studies. Additional details regarding the included articles can be found in Table 2.

### 3.2. Study Assessment: Methodological Quality, Psychometric Properties, and Quality of Evidence of the Scale

The most frequently reported psychometric properties were responsiveness (31 studies), internal consistency (20 studies), and hypothesis testing (14 studies). Those least frequently reported were content validity (1 study) and measurement error (1 study).

It should be noted that only eight of the studies were specifically designed and conducted to evaluate psychometric properties of the IMR-S. These eight psychometric studies provided information about different aspects of the scale, such as content validity (n = 1), internal consistency (n = 8), reliability (n = 3), hypothesis testing (n = 5), and responsiveness (n = 1). 

#### 3.2.1. Content Validity

Only one study (2.2%) evaluated content validity, that is, the degree to which the scale’s content accurately and comprehensively reflects the construct or concept it purports to measure; or to put it another way, whether the items in the instrument represent the full scope of the target construct accurately and comprehensively using qualitative and quantitative methods. This study was graded as being of doubtful methodological quality because it only provided information from clinicians regarding the relevance, comprehensiveness, and comprehensibility of the client version scale. Additionally, and because only a specific group of professionals (i.e., nurses) was consulted [53], the quality of evidence for content validity was graded as low. However, it should be noted that the IMR-S is based on the retrieval model, so although it is necessary to provide evidence of content validity in terms of understanding by users themselves, from the point of view of construct validity, this study provides sufficient evidence.

#### 3.2.2. Structural Validity

Three studies (6.5%) provided information about structural validity, which concerns the degree to which scale scores accurately represent the underlying dimension of the construct being measured. Of these, two studies [27,50] were rated as having very good methodological quality, since the internal structure of the IMR-S was analyzed by confirmatory factor analysis [27], and using several indices of model fit [50]. Both studies employed an adequate sample size for their analysis (n = 697 and n = 10,659, respectively). Finally, the results regarding the goodness of fit were found to be adequate. One of them [27] performed confirmatory factor analysis, while the other [50] applied the Rasch measurement model. The third study involved an exploratory factor analysis [45], and it was, therefore, considered adequate in terms of methodological quality. The quality of the overall rating is limited because the factor structure is not the same across studies. A three-factor structure was commonly reported for the IMR-S, corresponding to coping, recovery, and symptoms (F1), management and personal goals (F2), and effective use of medication and substance abuse (F3). For each of these subscales, the reported values of reliability (in terms of internal consistency) were acceptable. However, the results referring to F3 varied across the three studies; different factorial patterns were found, mainly due to homogeneity in responses to the items referring to the use of medications and substances. The quality of evidence was high, since it met all the requirements of the GRADE approach.

#### 3.2.3. Internal Consistency

Internal consistency, or the degree of interrelatedness among items, was calculated in 20 studies (43.5%) using Cronbach’s alpha. Most of these studies (n = 18) showed very good methodological quality, presenting alpha values for all scales and their factors, as indicated in COSMIN guidelines. Only two studies [18,50] were rated as doubtful, since they did not indicate alpha values for all of the IMR-S factors mentioned.

The client version of the IMR-S was analyzed in 18 studies. Of these, 11 studies (61.1%) reported internal consistency values greater than 0.70, indicating satisfactory internal consistency of the scores [22,24,59]. However, one study [40] reported an alpha value below 0.50, suggesting inadequate internal consistency [22,24]. The clinician version of the IMR-S was analyzed in nine studies, with eight (88.9%) reporting internal consistency values greater than 0.70. The other study [18] reported an alpha value of 0.69, slightly below the recommended threshold for satisfactory internal consistency [22]. 

The analysis of internal consistency by factors was reported in five studies. The coping, recovery, and symptoms factor (F1) yielded the highest values for both versions of the scale; the lowest values corresponded to the effective use of medication and substance abuse factor (F3), with only one study reporting values above 0.70 for the client version of the IMR-S [15].

Overall, therefore, the data for internal consistency suggest that the scale is consistent and useful for clinical application. It is important to note, however, that when each subscale is examined separately, some studies report lower internal consistency values. Table 3 displays the minimum and maximum internal consistency values reported for the entire scale and for each factor of the two versions of the IMR-S.

#### 3.2.4. Reliability

Reliability, defined as the proportion of the total variance in measurement that can be attributed to true differences between individuals, is usually evaluated through repeated measurements within a defined period, during which there has been no change in the individual’s condition. Reliability was examined in only four studies (8.7%) using repeated measures. Three studies calculated Pearson’s correlation coefficient, with values ranging from 0.64 to 0.88, while the other calculated the intraclass correlation coefficient, whose value was 0.71. Each of these studies was rated as methodologically adequate. However, the quality of their evidence was graded as moderate due to the small sample sizes (fewer than 100 participants in each case). The measurement properties were rated as indeterminate (?) in this domain, because relevant information such as weighted Kappa was not reported.

#### 3.2.5. Measurement Error

Measurement error, which refers to systematic and random error in an individual’s score (i.e., it is not attributable to true changes in the construct measured), was only analyzed in one study (2.2%), which calculated values for the standard error of measurement (SEM) and minimal important change (MIC) [30]. This study had adequate methodological quality, and the measurement properties were rated as sufficient (+), although the small sample size (n = 60) may have influenced the MIC value and its interpretability. The sample of this study was formed by 60 people from the Netherlands, diagnosed with psychotic disorder, mood anxiety disorders or other conditions, and varied in the level of daily support needed. The quality of evidence was rated as *low*, as only one study reported information about this psychometric characteristic, and the sample size was smaller than 100.

#### 3.2.6. Hypothesis Testing

Hypothesis testing, which concerns the degree to which scale scores align with the predicted hypotheses, provides information about convergent validity. This property was analyzed in 14 papers (30.4%). Some of these studies had a doubtful methodological quality, as important characteristics of the subgroups were not adequately described. Nonetheless, the measurement properties for hypothesis testing received a positive rating (+), because most of these studies reported results consistent with the hypotheses. Specifically, higher scores on the IMR-S correlated with lower scores on psychopathological symptom scales. Additionally, an increase in IMR-S scores was associated with improvement on scales measuring hope, social support, coping strategies, empowerment, symptom management, and control of life [21,44,45,47]. These findings further support the convergent validity of IMR-S scores for measuring recovery and their alignment with anticipated hypotheses. The samples of these studies were primarily composed of individuals with schizophrenia, bipolar disorder, anxiety, or major depression. The participants were mainly outpatients living independently. The quality of evidence was, however, moderate, due to the doubtful rating awarded to some studies [10,48].

#### 3.2.7. Responsiveness

Responsiveness, which refers to the ability of scale scores to detect changes in the construct over time, was the most widely analyzed property, in a total of 31 studies (67.4%). However, these studies primarily focused on assessing changes in recovery after implementing an intervention program. That is to say, the IMR-S was used to evaluate mean changes in participants, but the focus was not specifically on the ability of the scale to detect change. Data showed that 77.4% of studies detected a positive change effect in recovery, 19.4% did not detect any change, and 3.2% detected a negative change effect. A positive change effect suggests that IMR intervention produces improvements in the perception of the recovery process. Regarding methodological quality, 20 studies were rated as inadequate due to the interventions being poorly described and a lack of information about events occurring during the intervention period. In most studies, the reported results for post-treatment change were in accordance with the hypotheses, and consequently the quality of this psychometric property was rated as (+). The samples in these studies primary comprised individuals diagnosed with schizophrenia, bipolar disorder, anxiety, major depression, or personality disorder. Most participants were outpatients living independently; however, we identified three studies involving inpatient participants [6,16,52]. However, the quality of evidence was low, due to the inadequate rating awarded to some studies.

Table 4 summarizes quality levels for each measurement property.

## 4. Discussion

The principal aim of this systematic review was to assess the psychometric properties of the IMR-S so as to determine the reliability and suitability of its scores for evaluating recovery in patients with SMD. A broad and accurate assessment of this scale has not previously been conducted using the COSMIN Risk of Bias checklist [22], a methodology that allows the psychometric characteristics of a scale to be examined in a systematic and orderly way. COSMIN provides information about content, structural, transcultural, criterion, and convergent validity, and also about internal consistency, reliability, and responsiveness.

Over the years, the IMR-S has been frequently used alongside various recovery scales. Shanks et al. (2013) [61] conducted a systematic review of 12 different recovery measures available at that time, but they did not use the COSMIN methodology. The present review, therefore, adds to the literature by presenting a complete set of psychometric information for the IMR-S in a structured way.

A total of 46 studies, published between January 2004 and May 2023, were selected for analysis. Given that the concept of recovery focuses on the personal perception of the individual with a mental health problem [1], it is worth noting that 93% of these studies applied the client version of the scale, either independently or in combination with the clinician scale. For many years, people’s own perception of their recovery process was not taken into account when making decisions about treatment. In this respect, using a specific scale to capture their perspective is a step forward, insofar as it enhances their active participation in the recovery process. Furthermore, it aligns both with the principles outlined in the Convention on the Rights of Persons with Disabilities (CRPD, United Nations, 2006) [62] and with evolving mental health strategies adopted by numerous countries in recent years [63]. 

The methodological quality of the studies reviewed is insufficient in some COSMIN domains (boxes), such as content and structural validity, error measurement, and reliability. For these measures, few studies (<5) provide information on how they have been conducted. Content validity was analyzed in only one study, in which information on the client version of the scale was obtained from clinicians. Furthermore, the clinicians who assessed the scale were nurses, who usually apply intervention programs of a psycho-educational nature. This means that knowledge of recovery processes from other professional perspectives is still limited. Additionally, because client perceptions were not considered, more studies are needed to provide further evidence of content validity.

Structural validity was analyzed in three studies [27,45,50], each presenting a three-factor structure. Although these three studies agree on the overall structure of the IMR-S in terms of the number of factors, they differ regarding which items belong to each of the three factors. The different approaches to analyze the structural validity (Exploratory Factor Analysis [45], Confirmatory Factor Analysis [27], and Rasch measurement model [53]) might explain the heterogeneous findings in dimensionality.

Measurement error was analyzed in one study, which calculated the MIC value to evaluate the client perspective on change in scores. This information enables changes to be made to a client’s management through dialog between professionals and patients, as proposed by the recovery model. However, more studies are needed to bolster the results found in this research. Reliability was analyzed in four studies with adequate methodological quality. However, assessing reliability requires participants who are stable in the construct to be measured, and because recovery is a changing process, this criterion is difficult to fulfill. Therefore, assessing the reliability of the IMR-S will require short time intervals and comparison between groups of people at a similar stage of recovery. Information from the clinician version of the IMR-S can serve as a reference here and complement the perceptions of clients themselves.

Internal consistency was assessed in a significant number of studies, which showed satisfactory methodological quality. The client version of the IMR-S was the most commonly analyzed, and in most cases, reported values indicated consistency of scale scores. The clinician version presents the best internal consistency values. According to Bland criteria [64], the IMR-S achieves acceptable values of internal consistency when its total score is considered, being a satisfactory value for group comparison. However, the internal consistency values in studies that analyze the three-factor structure [15,18,27,45,47,50,53] and reported alpha values for each subscale were only moderate. Therefore, a more detailed analysis of individual scale items and of the factors or dimensions identified in previous studies [27,45] should be conducted. In this regard, it is worth noting that the use of medication or issues related to substance abuse are factors that have been identified as having a questionable impact on the perception of recovery [27], and items 13, 14, and 15 of the IMR-S are associated with these aspects. It is important to bear in mind here that historically, mental health problems have often been wrongly associated with drug use, leading to social stigma. The notion that drug use is closely linked to mental health issues may deter individuals from seeking the necessary help and support for their mental health issues [65,66]. 

Hypothesis testing was widely assessed in the studies reviewed, revealing significant correlations between the IMR-S and related constructs: hope, coping strategies, empowerment, symptom management, and control of life. All of these aspects are closely related to the CHIME recovery model [67], which includes factors that favor recovery: bond and supportive relationships, hope, personal identity, healthy coping strategies, and empowerment. Accordingly, IMR-S scores are shown to be related to essential concepts for describing an individual’s recovery.

The most extensively evaluated psychometric property was responsiveness. However, as previously mentioned, these studies did not specifically focus on the responsiveness of the IMR-S per se, but rather applied the scale to assess recovery after a psychoeducational intervention of some kind. This property (responsiveness), therefore, needs to be studied and analyzed in depth through specific studies, not least because the studies reviewed here were of doubtful or insufficient quality. Although the IMR-S was developed to evaluate recovery following the implementation of intervention programs [7], the reviewed studies do not provide sufficient information about the interventions carried out, and this hampers comprehensive analysis.

Although certain psychometric properties, such as content validity, require further investigation [12,40], the scale’s utility is evident due to its concise number of items, straightforward scoring method, incorporation of recovery model topics, and its capacity to capture the perspectives of both key stakeholders in the process (i.e., clients and clinicians). Incorporating the perspectives of clinicians and clients can facilitate interventions more closely aligned with clients’ needs. This enhances the understanding of the recovery process, leading to greater personal well-being and self-esteem improvements among clients. Meanwhile, clinicians may shift their recovery perspective by moving their focus from symptom reduction to emphasizing the developmental areas outlined in the CHIME model [3]. This makes the scale a valuable tool for assessing recovery in a global and practical way [11,12,37]. 

The eight psychometric studies included in this review exhibit heterogeneity in terms of the sociodemographic characteristics of the participants. Therefore, this diversity may limit the generalizability of the results concerning the psychometric properties of the IMR-S across different populations and recovery pathways.

Finally, a number of information gaps have been detected in this review. First, we identified no studies reporting information on criterion validity. Second, although different cultural adaptations have been identified, these studies do not provide detailed information about the adaptation procedure carried out or the psychometric properties of the adapted versions, nor do they provide evidence about cross-cultural metric equivalence [6,12,18,20,33,35,37,44]. 

A limitation of this systematic review is its exclusive inclusion of studies published in English. As a result, it may have overlooked certain psychometric properties of the IMR-S assessed in studies published in other languages. 

## 5. Conclusions

The IMR-S is a tool for measuring recovery in mental health from the client and clinician perspectives, and it shows a strong relationship with factors that favor the recovery of people with SMD, such as hope, control of life, identity, symptom management, and coping. The results of this systematic review suggest that the IMR-S is a valid and reliable (in terms of internal consistency) instrument that is suitable for use in clinical practice.

The use of scales of this kind can help to guide clinical practice by facilitating dialog between clients and clinicians and by empowering clients with SMD in decision making. 

However, caution is advised when it comes to interventions focused on medication use and substance abuse. Future research should assess the relevance of evaluating substance use and medication intake, suggesting possible modifications of the IMR-S items related to these factors, as they are less relevant within the framework of the person-centered recovery model. Additionally, studies should explore the divergence and convergence between clinician and client perceptions of recovery by analyzing differences and similarities in recovery scores. This analysis could provide insights into the shared or divergent understandings of the recovery process. Furthermore, future investigations should examine how insights gained from the scale can influence clinical decision-making and treatment planning, potentially improving recovery outcomes.

Further research is required to develop a fuller understanding of the scale’s psychometric properties, especially as regards those aspects about which less is currently known, such as criterion validity, content validity, and measurement error.

## Figures and Tables

**Figure 1 behavsci-14-00340-f001:**
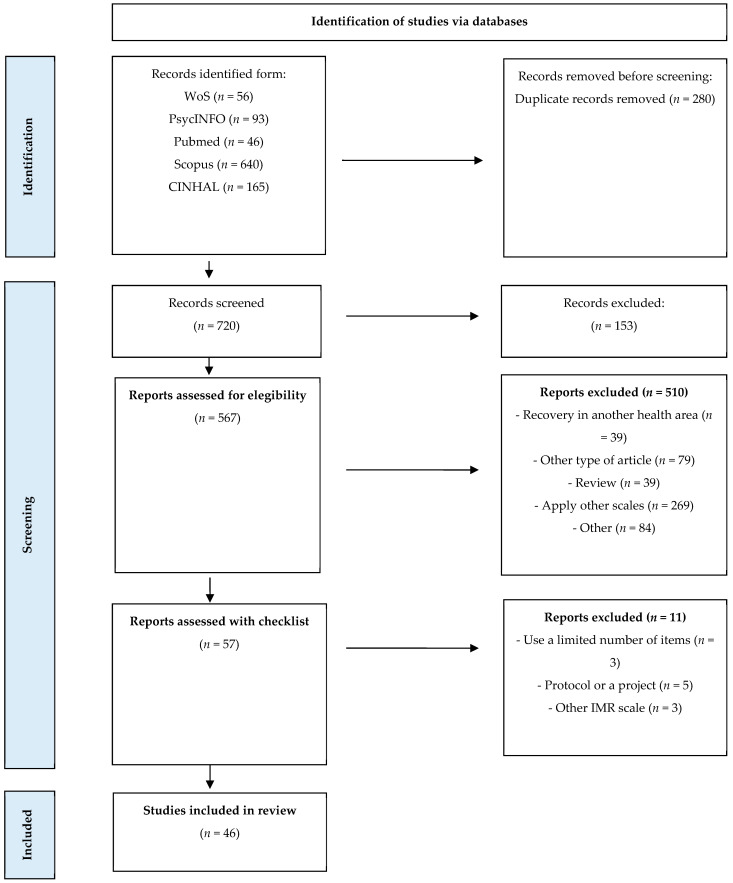
PRISMA flow chart.

**Table 1 behavsci-14-00340-t001:** COSMIN Risk of Bias checklist summary [22].

Content Validity	
**Box 1**	Outcome measure tool development
**Box 2**	Content validity
**Internal Structure**	
**Box 3**	Structural validity
**Box 4**	Internal consistency
**Box 5**	Cross-cultural validity/measurement invariance
**Remaining Measurement Properties**	
**Box 6**	Reliability
**Box 7**	Measurement error
**Box 8**	Criterion validity
**Box 9**	Hypothesis testing for construct validity
**Box 10**	Responsiveness

**Table 2 behavsci-14-00340-t002:** General characteristics of articles included in the study.

Authors	Sample Size	Females n (%)	Mean Sample Age	IMR-S	
				Version Applied	No. of Items
Barbic et al., 2018 ^b^ [21]	228	112 (49.1%)	45.8	Client	13
Bartels et al., 2014 ^b^ [28]	71	39 (43.7%)	60.3	Both	15
Beentjes et al., 2018 ^b^ [29]	60	36 (60.0%)	43.8	Client	15
Beentjes et al., 2021 ^b^ [30]	60	36 (60.0%)	45	Client	15
Bendel-Rozow, 2021 ^b^ [4]	52	13 (25.0%)	48	Both	15
Ben-Zeev et al., 2020 ^b^ [31]	49	22 (44.9%)	44.8	Client	15
Buck et al., 2022 ^b^ [32]	17	5 (29.4%)	55.1	Client	15
Casey et al., 2023 ^b^ [33]	23	7 (30.4%)	35	Client	15
Chinman et al., 2015 ^b^ [34]	238	28 (11.8%)	53.2	Client	15
Daas-Iraqi et al., 2020 ^b^ [35]	150	NS	40.9	Both	15
Dalum et al., 2016 ^b^ [18]	198	89 (44.9%)	43	Both	15
Egeland et al., 2017 ^b^ [20]	44	16 (36.4%)	40.7	Both	15
Färdig et al., 2011 ^b^ [36]	102	41 (38.3%)	43	Both	15
Färdig et al., 2011 ^b^ [37]	41	19 (46.3%)	40.4	Both	15
Färdig et al., 2016 ^b^ [38]	53	23 (43.4%)	41.5	Both	15
Firmin et al., 2015 ^b^ [39]	46	11 (23.9%)	48.5	Client	15
Fortuna et al., 2018 ^b^ [26]	8	7 (87.5%)	68.8	Client	15
Fortuna et al., 2020 ^b^ [40]	341	NS	NS	Client	15
Fortuna et al., 2022 ^b^ [41]	21	15 (71.4%)	39.9	Client	15
Garber-Epstein et al., 2013 ^b^ [42]	252	112 (44.4%)	43.5	Client	13
Gilmer et al., 2016 ^b^ [43]	1279	NS	NS	Clinician	15
Goosens et al., 2017 ^a^ [44]	111	53 (47.7%)	Between 18 →65	Both	15
Hasson-Ohayon et al., 2007 ^a^ [15]	210	73 (34.8%)	34.7	Both	15
Hasson-Ohayon et al., 2008 ^a^ [45]	210	73 (34.8%)	34.6	Both	15
Jensen et al., 2019 ^b^ [46]	198	89 (44.9%)	43	Both	15
Kukla et al., 2013 ^b^ [47]	119	25 (21.0%)	47.6	Both	15
Levitt et al., 2009 ^b^ [16]	104	38 (36.5%)	53.9	Both	15
MacMillan et al., 2021 ^a^ [10]	77	58 (75.3%)	Between 18 → 50	Client	14
Matthias et al., 2017 ^b^ [48]	63	9 (14.3%)	53	Client	15
Matthias et al., 2014 ^b^ [49]	63	9 (14.3%)	53	Client	15
McGuire et al., 2014 ^a^ [50]	697	331 (47.5%)	NS	Clinician	15
McGuire et al., 2017 ^b^ [51]	236	97 (41.1%)	45.2	Client	15
Miyajima et al., 2023 ^b^ [52]	10	7 (70.0%)	Median 50.5	Both	15
Polat &Kutlu 2021 ^b^ [6]	50	26 (52%)	39.1	Client	15
Polat et al., 2020 ^a^ [53]	75	30 (40.0%)	41.6	Client	15
Pratt et al., 2011 ^b^ [54]	44	29 (65.9%)	46.4	Client	15
Pratt et al., 2015 ^b^ [55]	38	27 (71.0%)	46.4	Client	15
Roosenchoon et al., 2016 ^b^ [19]	61	33 (54.1%)	42.4	Both	15
Roosenchoon et al., 2021 ^b^ [3]	187	88 (47.1%)	44.2	Both	15
Salyers et al., 2007 ^a^ [56]	59	20 (33.9%)	43.5	Both	15
Salyers et al., 2009 ^b^ [57]	40	21 (52.5%)	43.5	Client	15
Salyers et al., 2010 ^b^ [58]	324	148 (45.7%)	42.3	Both	15
Salyers et al., 2014 ^b^ [17]	118	24 (20.3%)	47.7	Client	15
Sklar et al., 2012 ^a^ [27]	10659	5283 (49.6%)	43.2	Clinician	15
Tan et al., 2017 ^b^ [59]	50	31 (62.0%)	44.2	Both	15
White et al., 2017 ^b^ [60]	236	97 (41.1%)	45.2	Client	15

^a^: psychometric study; ^b^: other type of study (clinical intervention, case–control, exploratory, etc.). NS = not specified.

**Table 3 behavsci-14-00340-t003:** Internal consistency values.

		IMR-S Client Version	IMR-S Clinician Version
Total Scale	Max	0.85 [59]	0.85 [59]
	Min	0.49 [40]	0.69 [18]
F1: Coping, recovery, and symptoms	Max	0.73 [15]	0.83 [15]
	Min	0.69 [44,53]	0.67 [50]
F2: Management and personal goals	Max	0.69 [53]	0.83 [27]
	Min	0.50 [15]	0.61 [44]
F3: Effective use of medication and substance abuse	Max	0.74 [15]	0.69 [27]
	Min	0.35 [53]	0.28 [44]

F = Factor.

**Table 4 behavsci-14-00340-t004:** Summary of measurement properties of the IMR-S.

Measurement Property	COSMIN Risk of Bias Checklist Ratings				Quality of Evidence
	Very Good	Adequate	Doubtful	Inadequate	
Content validity (n = 1)	0	0	1	0	Low
Structural validity (n = 3)	2	1	0	0	High (?)
Internal consistency (n = 20)	18	0	2	0	Moderate (−)
Reliability (n = 4)	0	4	0	0	Moderate (?)
Measurement error (n = 1)	0	1	0	0	Low (+)
Hypothesis testing (n = 14)	8	4	2	0	Moderate (+)
Responsiveness (n = 31)	0	0	11	20	Low (+)

n = number of studies assessing the measurement property.

## Data Availability

Not applicable.

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
