# Peer review of "Assessing the Psychometric Properties of the Illness Management and Recovery Scale: A Systematic Review Using the Consensus-Based Standards for the Selection of Health Measurement Instruments (COSMIN)"

_behavsci, 2024, doi:10.3390/bs14040340_

Round 1
Reviewer 1 Report (New Reviewer)
Comments and Suggestions for Authors
Author Response
Thank you for inviting us to resubmit our manuscript, titled “Assessing the psychometric properties of the Illness Management and Recovery Scale: a systematic review using the Consensus-based Standards for the Selection of Health Measurement INstruments (COSMIN)”, to Behavioral Sciences following revision.
My co-authors and I have carefully reviewed the manuscript in light of the reviewers’ comments and suggestions. All added or reformulated information has now been highlighted in yellow. We believe the clarifications requested significantly contribute to enhancing the quality of our study. We hope we have addressed each comment comprehensively. We are delighted to have the opportunity to publish our work in Behavioral Sciences.
The manuscript under review undertakes a comprehensive evaluation of the Illness Management and Recovery Scale (IMR-S) by conducting a systematic review of its psychometric properties in alignment with the Consensus-Based Standards for the Selection of Health Measurement Instruments (COSMIN). The study systematically searched five databases for relevant studies published from January 2004 to May 2023, ultimately including 46 papers in its analysis. This work is highly relevant to the field of Behavioral Sciences, especially concerning the assessment and recovery processes for individuals with severe mental disorders, aligning well with the journal's thematic focus.
Response: We appreciate the time you took to review our work and your constructive criticism. Below we have sought to address each of your comments.
- Abstract and Keywords: The abstract succinctly presents the study's aim, methods, findings, and implications. However, it lacks details regarding the study's limitations and future research directions. The chosen keywords are appropriate and cover the manuscript's core themes.
Response: Thank you for pointing this out. We have added the suggested information (page 1, lines 28-31).
- Background: The background section effectively sets the stage for the study by highlighting the significance of the recovery model in mental health and introducing the IMR program. Nonetheless, it could benefit from a more detailed discussion on the IMR-S's development context and its operationalization of recovery concepts.
Response: Thank you for pointing this out. We have added the suggested information (page 2, lines 55-59).
- Methods: The methodology is generally well-structured, following the COSMIN checklist and PRISMA guidelines. The search strategy, inclusion/exclusion criteria, and study selection process are adequately detailed. However, the manuscript could improve by providing more information on the data synthesis approach and addressing potential biases in study selection and data extraction.
Response: Thank you for insightful comments regarding the Methods section of our paper. Based on your feedback, we have added some information and clarified the possible biases of this systematic review (page 3, lines 123 - 125)
- Results: The results section comprehensively presents the findings from the included studies, covering various psychometric properties of the IMR-S. While the presentation of results is clear, the section could be enhanced by integrating tables or figures summarizing the key findings, especially regarding the methodological quality assessments and the psychometric properties evaluated.
Response: Thanks for your comment. Some changes have been made to improve the way information regarding methodological quality assessments and the evaluation of psychometric properties is summarized (page 10, line 327). Additionally, more detailed information about the data extraction from each box in the different studies is shown in the supplementary material.
- Discussion: The discussion effectively interprets the findings, contextualizing them within the broader literature on recovery measurement tools. The authors acknowledge the strengths and limitations of the IMR-S and the current study, but the discussion could be deepened by exploring the implications of these findings for clinical practice and research more thoroughly.
Response: Thank you for highlighting this. Based on your suggestion, we have delved into the implications of using this scale on clients' perceptions and clinical practice (pages 12 and 13, lines 416-421).
- Conclusion: The conclusion succinctly summarizes the study's contributions and underscores the IMR-S's utility in clinical and research settings. It appropriately calls for further research to address identified gaps. However, it could be more forceful in suggesting specific directions for future studies or potential modifications to the IMR-S.
Response: Thank you for this observation. We have extended our conclusion, focusing on future research directions (page 13, lines 446-454)
Summary and Recommendation Overall, this manuscript makes a valuable contribution to the understanding of the IMR-S's psychometric properties and its application in assessing recovery in individuals with severe mental disorders. While it demonstrates a rigorous approach to systematic review, there are areas where further elaboration and clarification could enhance its impact. Considering the significance of the findings and the relatively minor concerns raised, I recommend Minor Revisions to address the identified issues. With these revisions, the manuscript would be a strong fit for publication in the Behavioral Sciences journal.
We appreciate the time you took to review our work and your constructive criticism. We have sought to address each of your comments.
Reviewer 2 Report (New Reviewer)
Comments and Suggestions for Authors
The IMR program aims to enhance self-management strategies and collaboration between clinicians and individuals. Eligible studies underwent assessment for general characteristics and methodological quality. The Illness Management and Recovery Scale (IMR-S) assesses recovery in severe mental disorders. This study conducted a systematic review of 46 papers to analyze its psychometric properties, confirming the efficiency of IMR-S with data from these studies. However, further research is warranted to explore validity and internal factor structure. Although responsiveness was extensively evaluated, more focused studies are necessary. The manuscript is well-written, and the IMR-S approach holds promise due to its concise items and ability to capture stakeholder perspectives. I recommend accepting this manuscript for publication with the following minor improvements:
- As the author mentioned, exploring applications in languages other than English would complement the efficacy of this approach significantly.
- Providing more comparisons with current approaches to demonstrate the advantages of this approach would enhance the manuscript. Although challenging, any comparisons, both in approach and results, would be intriguing.
- A detailed discussion of the results and advantages of this approach would add value. It would be engaging for the author to discuss the approach's benefits and its potential future applications.
Author Response
Thank you for inviting us to resubmit our manuscript, titled “Assessing the psychometric properties of the Illness Management and Recovery Scale: a systematic review using the Consensus-based Standards for the Selection of Health Measurement INstruments (COSMIN)”, to Behavioral Sciences following revision.
My co-authors and I have carefully reviewed the manuscript in light of the reviewers’ comments and suggestions. All added or reformulated information has now been highlighted in yellow. We believe the clarifications requested significantly contribute to enhancing the quality of our study. We hope we have addressed each comment comprehensively. We are delighted to have the opportunity to publish our work in Behavioral Sciences.
The IMR program aims to enhance self-management strategies and collaboration between clinicians and individuals. Eligible studies underwent assessment for general characteristics and methodological quality. The Illness Management and Recovery Scale (IMR-S) assesses recovery in severe mental disorders. This study conducted a systematic review of 46 papers to analyze its psychometric properties, confirming the efficiency of IMR-S with data from these studies. However, further research is warranted to explore validity and internal factor structure. Although responsiveness was extensively evaluated, more focused studies are necessary. The manuscript is well-written, and the IMR-S approach holds promise due to its concise items and ability to capture stakeholder perspectives. I recommend accepting this manuscript for publication with the following minor improvements:
Thank you very much for taking the time to review this manuscript. Please find the detailed responses below and the corresponding revisions highlighted in the re-submitted files.
- As the author mentioned, exploring applications in languages other than English would complement the efficacy of this approach significantly.
Response:We agree that limiting the search to studies published in English may have resulted in the omission of some publications. Therefore, we have added a comment highlighting that a limitation of the study is having restricted the search for publications exclusively to those published in English(page 1, lines 28-31 and page 13, lines 431-433).
- Providing more comparisons with current approaches to demonstrate the advantages of this approach would enhance the manuscript. Although challenging, any comparisons, both in approach and results, would be intriguing.
Response: Thank you for your comment. In the new version of the manuscript, we delve deeper into the implications of applying the scale to assess clients' perceptions and clinical practice (page 13, lines 446-454).
- A detailed discussion of the results and advantages of this approach would add value. It would be engaging for the author to discuss the approach's benefits and its potential future applications.
Response: Thank you for this valuable feedback. We have added some information on the clinical practice relevance of using this scale. (page 12 and 13, lines 416-421)
Reviewer 3 Report (New Reviewer)
Comments and Suggestions for Authors
This manuscript is worthy of publication. I have no suggestions for improvement. Good Luck.
Author Response
This manuscript is worthy of publication. I have no suggestions for improvement. Good Luc k.
Response: We would like to thank Reviewer 3 for his/her positive comments.
Reviewer 4 Report (New Reviewer)
Comments and Suggestions for Authors
While the study is well-designed and executed, there are several issues that need to be addressed:
1. coherence between objective and methods that it stated principal aim of assessing the psychometric properties to determine the reliability and suitability of the IMR-S for evaluating recovery is not directly addressed by the method employed. I think the author should either align with the COSMIN methodology used or expand the method section to describe how the findings from the 3 cosmin phases will be used to achieve the stated objective of this review.
2. maybe author could consider conducting subgroup analyses to examine the psychometric properties within more homogenous subsets of studies based on relevent characteristics such as diagnosis, intervention type or culture context.
3. in the result and discussion section, author can provide clear guidance on the population and contexts in which the IMR-s has demonstrated satisfactory psychometric properties based on their findings.
4. to explicitly point out its applicability / generability of findings, that this review has significant heterogeniety in terms of participants demographic, etc and the diversity raise concerns about the generability of the findings on the IMR-s psychometric properties across different population n recovery pathways.
5. in addition, please acknowledge the potential impact on the interpretation of the results
6. What is your recommendation ?
Author Response
Thank you for inviting us to resubmit our manuscript, titled “Assessing the psychometric properties of the Illness Management and Recovery Scale: a systematic review using the Consensus-based Standards for the Selection of Health Measurement INstruments (COSMIN)”, to Behavioral Sciences following revision.
My co-authors and I have carefully reviewed the manuscript in light of the reviewers’ comments and suggestions. All added or reformulated information has now been highlighted in yellow. We believe the clarifications requested significantly contribute to enhancing the quality of our study. We hope we have addressed each comment comprehensively. We are delighted to have the opportunity to publish our work in Behavioral Sciences.
While the study is well-designed and executed, there are several issues that need to be addressed:
coherence between objective and methods that it stated principal aim of assessing the psychometric properties to determine the reliability and suitability of the IMR-S for evaluating recovery is not directly addressed by the method employed. I think the author should either align with the COSMIN methodology used or expand the method section to describe how the findings from the 3 cosmin phases will be used to achieve the stated objective of this review.
Response: Thank you for this suggestion. In the new version of the manuscript, we have attempted to improve this by making various changes throughout the manuscrit (page 2, lines 72, 73-77 and 81-82; page 4, lines 143-144; 150-151 and 159-160).
- maybe author could consider conducting subgroup analyses to examine the psychometric properties within more homogenous subsets of studies based on relevent characteristics such as diagnosis, intervention type or culture context.
Response: Thank you very much for your feedback. Unfortunately, the limited number of strictly psychometric studies identified does not allow for subgroup analyses based on relevant characteristics such as diagnosis, intervention type, or cultural context. However, this approach should be considered for future research on this instrument.
- in the result and discussion section, author can provide clear guidance on the population and contexts in which the IMR-s has demonstrated satisfactory psychometric properties based on their findings.
Response: Thank you for highlighting this. We have included additional information about the sample type within the sections discussing psychometric properties (page 9, lines 283-285; and page 10, lines 300-302 and 318-322).
4. to explicitly point out its applicability / generability of findings, that this review has significant heterogeniety in terms of participants demographic, etc and the diversity raise concerns about the generability of the findings on the IMR-s psychometric properties across different population n recovery pathways.
Response: Thank you for your comment. We have included information regarding study’s limitations in terms of generability (page 12, lines 423-426).
5. in addition, please acknowledge the potential impact on the interpretation of the results
Response: Thank you for this observation. We have extended our conclusion, focusing on future research directions (page 13, lines 446-454)
- What is your recommendation ?
Response: Based on our results and considering that the psychometric properties of the IMR-S have been reported from a wide range of studies classified with good methodology, we believe the scale is capable of demonstrating its potential to offer guidance for clinical practice. Our recommendation has been incorporated in the abstract and in the conclusion (page 1, lines 28-31 and page 13, lines 446-454).
Round 2
Reviewer 4 Report (New Reviewer)
Comments and Suggestions for Authors
The authors committed in revising the manuscript and addressed all the issues I mentioned in the review report. I have no further comments.
This manuscript is a resubmission of an earlier submission. The following is a list of the peer review reports and author responses from that submission.
Round 1
Reviewer 1 Report
Comments and Suggestions for Authors
Summary:
This study aimed to assess the psychometric properties of the IMR-S for evaluating recovery. Two coders searched five databases for relevant studies published between 2004 and 2023. The COSMIN Risk of Bias checklist was used for assessment, resulting in 46 included papers. Most studies showed high internal validity, while responsiveness was limited. Convergent validity and measurement error were positive, and structural validity studies had high-quality evidence. The IMR-S is widely used to assess post-treatment changes, offering valuable guidance for clinical practice.
Comments:
· In Section 3.1, it is imperative to provide a comprehensive rationale for the exclusion of these studies, given the substantial reduction of data by 95%. It is particularly essential to elucidate the intricacies of the PRISMA flow chart, including the methodology employed for identifying duplicate papers. It should be emphasized that the presence of duplicate papers raises concerns about their publication validity.
· In Section 3.1, please regenerate the PRISMA flow chart, ensuring that the identification section reads: "studies identified through…."
· Regarding Table 2 in Section 3.1, the information presented in the authors' column appears to be unclear and confusing. It may require clarification or reformatting to enhance its comprehensibility.
· In Section 3.2.2 (line 214) and Section 3.2.3 (line 229), instead of using the term "very good," it is essential to specify the precise reasons why the model is superior. Please outline the comparative advantages and strengths of the model.
· In section 3.2.3 (line 236), what is internal consistency value greater than 0.70 indicating? What is alpha value of 0.69 means?
· In Section 3.2.6 (line 284), please add a citation at the end of the paragraph to provide a source for the information presented. Additionally, specify which studies are being referred to as "some studies" to enhance clarity and transparency.
Reviewer 2 Report
Comments and Suggestions for Authors
This paper conducted a systematic review evaluating the psychometric properties of the Illness Management and Recovery Scale (IMR-S) using the Consensus-based Standards for the selection of health Measurement Instruments (COSMIN). The authors selected 46 articles from five databases, spanning from January 2004 to May 2023, and rated these articles on their psychometric qualities. It is important to focus on the psychometric qualities of IMR-S across studies and samples, yet there are several key issues that should be addressed before its publication.
1. Introduction. Given that the key focus of this review is on the psychometric properties across studies alone, the Introduction section is ill-organized. The introduction disproportionately focuses on MDS and 'recovery' definitions, diverting from the study's primary focus. On the flip side, as to why investigating the psychometric properties of IMR-S is so important that warrants a single review paper, the justification in the Introduction is inadequate. We are only told that the scale needs refining and it has not yet been examined by COSMIN (Lines 81-95). These are not sufficient grounds for a separate review. And given the current Introduction, it strikes me as only a side issue that should have been included as a subsection in a larger review on the effectiveness of IMR program.
2. Expand on IMR-S's basics, practical use, and issues, and provide a detailed introduction to COSMIN in both the Introduction and Methodology sections.
3. The main research method, the COSMIN, warrants a more detailed introduction, both in Introduction and in Methodology sections.
4. Echoing the first point, a crucial issue facing this paper is that the research goal is not sufficiently established. This manifests also in the discussion section. It lacks a discussion on the practical and theoretical implications of the study. I.e., why is it so important to rate the psychometric properties of IMR-S in the first place, since the main interest of the authors are seemingly neither in comparing different but similar scales, nor in psychometric techniques that could potentially improving the IMR-S?
Reviewer 3 Report
Comments and Suggestions for Authors
This paper presents a very interesting purpose, as it intends to verify the quality of an instrument for clinical practice with people suffering from severe mental disorders, which is a frequent and important problem that deserves special attention.
The text is very well organized and written, with coherence between the several sections. No spelling or grammatical errors were found.
The abstract presents a good synthesis of the whole text.
The background is well appropriate and this section ends with the aims of the study.
The methodology is clear (a systematic review) presenting the PROSPERO reference and all the needed requisites for this kind of research.
Results start with the PRISMA flow chart, which clarifies the process and the data obtained. Only on the identification step, in some boxes, it is not possible to see the complete phrase, as it is cut. Some graphical improvements are suggested here. Following, some attention should be given to the age range of the studies because in the text authors refer to “from 34 to 68.8 years” (page 5, line 170), but after, on table 2 (page 6) we can read on the column of Mean Sample Age “18 - 65” and “Between 18->50”. Thus, a revision or clarification is suggested.
The following results are elucidative and the discussion is presented in a good manner. Some limitations are also indicated in its final.
The text ends with conclusions about the quality of the IMR-S and some recommendations.
References are appropriate and according to the aims of the study.
It is a good paper to be published in the journal Behavioral Sciences.